# *Coxiella burnetii* Seroprevalence and Associated Risk Factors in Cattle, Sheep, and Goats in Estonia

**DOI:** 10.3390/microorganisms11040819

**Published:** 2023-03-23

**Authors:** Kädi Neare, Lea Tummeleht, Brian Lassen, Arvo Viltrop

**Affiliations:** 1Chair of Veterinary Biomedicine and Food Hygiene, Institute of Veterinary Medicine and Animal Sciences, Estonian University of Life Sciences, 51006 Tartu, Estonia; 2Research Group for Foodborne Pathogens and Epidemiology, National Food Institute, Technical University of Denmark, 2800 Kongens Lyngby, Denmark

**Keywords:** foodborne zoonotic diseases, infectious disease, one health, public health, vector-borne diseases

## Abstract

Q fever, a disease caused by *Coxiella burnetii* (*CB*), is an emerging zoonotic health problem. The prevalence data from potential sources are valuable for assessing the risk to human and animal health. To estimate the prevalence of *CB* antibodies in Estonian ruminants, pooled milk and serum samples from cattle (*Bos taurus*) and pooled serum samples from sheep (*Ovis aries*) and goats (*Capra hircus*) were analyzed. Additionally, bulk tank milk samples (BTM; n = 72) were analyzed for the presence of *CB* DNA. Questionnaires and herd-level datasets were used to identify the risk factors for exposure using binary logistic regression analysis. The prevalence of *CB*-positive dairy cattle herds (27.16%) was significantly higher than that in beef cattle herds (6.67%) and sheep flocks (2.35%). No *CB* antibodies were detected in the goat flocks. *CB* DNA was found in 11.36% of the BTM samples. The odds of seropositivity were higher in dairy cattle herds, with an increasing number of cattle in the herd, and with location in southwestern, northeastern and northwestern Estonia. Dairy cattle herds had higher odds of testing positive for *CB* in BTM if the dairy cows were kept loose and lower odds if the herd was located in northwestern Estonia.

## 1. Introduction

Q fever (QF) is a zoonosis caused by the intracellular bacterium *Coxiella burnetii* (*CB*). The bacteria are considered to be globally distributed but may be overlooked in some countries due to the irregular or absence of investigations [1,2]. The main transmission routes of *CB* are via the inhalation of spore-like stages transported in aerosols or via contact with excretions (urine, feces, vaginal mucus, semen, and milk) from infected animals or contaminated materials [3]. In rare cases, *CB* is transmitted through tick bites [4]. Humans can acquire infection by consuming unpasteurized milk or dairy products containing viable *CB* [3].

In ruminants, cats, and dogs, the course of the infection is commonly asymptomatic but can, in some cases, affect reproduction, manifesting as stillbirths, infertility, retained placenta, abortions, and weak offspring [5]. Domestic ruminants are considered the most important reservoir for human infections [3].

In humans, the average incubation period of *CB* infection is 2 to 3 weeks, and the symptoms include flu-like illness, pneumonia, and hepatitis. Chronic QF is rare and generally manifests clinically as vascular infection and endocarditis [3].

In response to the rising concerns about health risks regarding a QF outbreak in the Netherlands, the European Food Safety Authority (EFSA) assembled a scientific opinion that called for the need for harmonized passive and active monitoring of QF [1]. Following a Dutch outbreak, human *CB* infections were generally found near dairy goat farms with abortion waves induced by *CB* [6].

Herd-level seroprevalence estimates of *CB* infections in European ruminants range from 0.0% to 81.6% in cattle, 0.0% to 78.6% in sheep and 0.0% to 43.1% in goats [7,8,9,10]. In previous studies, herd size, a loose housing system, and location have been identified as risk factors for herd-level *CB* seropositivity. Breed, parity, use of windshield and use of artificial insemination were detected as animal-level risk factors [7,9,11].

In Estonia, there are few data on the presence of *CB* in domestic animals. Evidence of infection has previously been detected serologically in five cattle from three different farms [12]. Information is lacking on the presence of *CB* in other domestic animals, including sheep, goats, and pets.

This study aimed to estimate the exposure to *CB* in domestic ruminants in Estonia based on seroprevalence and identify potential associated risk factors.

## 2. Materials and Methods

### 2.1. Setting and Sample Size

#### 2.1.1. Estimation of CB Herd-Level Seroprevalence in Cattle

The study population included cattle herds sampled for the official surveillance of enzootic bovine leukosis (EBL) in Estonia. Milk or serum samples from all cattle aged ≥ 24 months were collected from all cattle herds in Estonia by authorized veterinarians for EBL surveillance in 2012. Available aliquots of these samples were stored at the Estonian National Centre for Laboratory Research and Risk Assessment and tested for *CB* antibodies at the Estonian University of Life Sciences (EMU).

In 2012, 4665 cattle herds were registered in the Estonian national production animal register [13] The herds that kept more dairy breed cattle than beef breeds were classified as dairy herds, and the remainder were considered as beef herds. Based on this categorization, there were 3417 dairy and 1248 beef herds.

Herds with at least five adult animals were included. Using a random number generator EpiTools^®^ [14], 504 cattle herds (337 dairy and 167 beef herds) were selected from the original list of herds and included in this study. The selection was further stratified by administrative region (county) in proportion to the number of herds in each herd category (dairy or beef) in each county.

Assuming a 20% herd prevalence in the population, the sample size enabled the estimation of the prevalence at the country level in each herd category, with 95% confidence and 5% accepted error.

From each beef herd, 30 serum samples were randomly selected for *CB* antibody testing, unless ≤30 samples were collected, in which case all the samples were tested. The selected serum samples were pooled from 3 to 6 samples before testing for *CB* antibodies using 150 µL of each sample.

Fifty individual milk samples were randomly selected from each dairy herd and pooled for *CB* antibody testing. If the number of individual samples per herd was ≤50, all the samples were pooled and tested.

The sampling scheme of included cattle herds is presented in Appendix A.

A sample size of 30 animals per herd enabled the detection of a seropositive herd with 95% confidence if the within-herd seroprevalence was 12%, assuming a sensitivity of 82.6% and a specificity of 100% for testing antibodies in cattle blood serum [15], and a very good match (Kappa = 0.89) between the results of serum and milk samples [16].

#### 2.1.2. Detection of CB Seropositive Sheep and Goat Flocks

In Estonia, 1949 sheep flocks and 653 goat flocks were registered in the production animal register in 2012 [13].

Flocks involved in the national breeding program and subjected to official surveillance for brucellosis were included. All herds sampled in 2012 and 2013 were included. Leftover serum samples from the *Brucella* test were used for further investigation of *CB* antibodies at EMU.

Additional sheep and goat flocks not included in the brucellosis surveillance program were sampled to increase the herd sample size. Blood samples were collected from 30 randomly selected adult animals from each flock.

The sampling procedure is summarized in Appendix A.

The number of flocks tested enabled the detection of an infected flock with 95% confidence if the herd-level prevalence in the population was 2% in sheep and 20% in goats, assuming herd-level sensitivity and specificity of 95% and 100%, respectively. The number of samples collected from each flock (1–65) enabled the identification of an infected flock with 95% confidence, at least on a 20% prevalence level, assuming 88.8% sensitivity and 98.5% specificity of the test in sheep, and 91.6% sensitivity and 98.9% specificity in goats, respectively [15].

#### 2.1.3. Risk Factor Analyses in Cattle Herds

Two studies were performed to identify risk factors for *CB* infection in Estonian cattle herds.

In the first study (Risk Factor Study 1), herds and data from the seroprevalence study based on surveillance program samples were included. Data on the herds’ geographical locations, sizes (number of animals), and breeds were retrieved from the National Animal Register database [13].

The second study (Risk Factor Study 2) included dairy cattle herds that volunteered to participate in the study in 2013. One tank milk sample (BTM) was submitted by farmers from each herd to test for *CB* antibodies and define the herd’s infection status (positive or negative), as specified in Appendix A. In addition, information on putative risk factors in the farm was gathered using a structured questionnaire (Appendix A).

### 2.2. Sample Analysis

Collected samples were stored at −20 °C in labelled plastic vials until analysis. All laboratory analyses were performed in 2012–2014.

#### 2.2.1. Antibody Detection

An indirect enzyme-linked immunosorbent assay PrioCHECK Ruminant Q Fever Ab Plate Kit (Thermo Fisher Scientific, Waltham, MA, USA) was used to detect *CB* antibodies in milk and serum samples. Sample preparation, analyses, and interpretation of the results were performed according to the manufacturer’s instructions [17].

#### 2.2.2. DNA Detection

All pooled milk samples that were positive for *CB* antibodies (n = 88) were tested for the presence of *CB* DNA using polymerase chain reaction. DNA was purified from milk samples using Chelex 100 Resin (Bio-Rad Laboratories, Hercules, CA, USA). Trans-1 (5′-TAT GTA TCC ACC GTA GCC AGT C-3′) and Trans-2 (5′-CCC AAC AAC ACC TCC TTA TTC-3′) primers were used to detect the presence of specific IS*1111* repetitive elements in *CB* DNA. The use of these primers had been described by Berri et al. [18]. Positive control was obtained from Vircell Microbiologists [19] and purified distilled water was used as no template negative control. Initial denaturation took place at 95 °C, the remaining denaturations at 94 °C. During first five cycles the annealing was performed at 66–62 °C (lowered 1 °C with each cycle), and the rest of the annealings were performed at 61 °C. Elongation processes were carried out at 72 °C and the reaction was terminated at 4 °C [20].

### 2.3. Statistical Analysis

#### 2.3.1. Prevalence Estimation

Cattle herds were categorized as dairy herds (≥50% dairy breed cattle) and beef herds (<50% dairy breed cattle), and herd-level prevalence estimates were calculated for both subgroups as well as for sheep and goat herds. Wilson’s 95% confidence intervals (CI) were calculated using the EpiTools^®^ epidemiological calculator ’Confidence limits for a proportion’ [14].

#### 2.3.2. Risk Factor Analysis

Logistic regression analyses were used to estimate the association between herd *CB* infection status based on antibodies and potential risk factors.

Herd size was evaluated in the models as a continuous variable (number of animals in the herd) and categorical variable (herd size). Prior analyses of cattle herds were categorized according to the number of animals as small (<101 animals), medium-sized (101–300 animals), and large herds (>300 animals).

The effects of specific cattle breeds (Estonian Holstein Friesian, Estonian Red, Blonde d’Aquitaine and Hereford) were evaluated using the number of animals of specific breeds registered in the herd.

Cattle herds were grouped into regions based on Estonian counties (N = 15): southwest (Pärnu, Saare, and Viljandi counties), southeast (Põlva, Tartu, Valga, and Võru counties), northeast (Ida-Viru, Jõgeva, Järva, and Lääne-Viru counties), and northwest (Harju, Hiiu, Lääne, and Rapla counties) (Appendix A).

The season of sample collection was determined as astronomical seasons: spring, summer, autumn, and winter, with the respective starting dates.

The percentage of registered beef cattle was calculated from the total number of animals registered in the cattle herd.

The keeping systems of lactating cows and pregnant heifers were categorized as tied, loose, or mixed (tied and loose).

The grazing of adult animals was categorized as ’no grazing’, ‘grazing of dry cows’, or ‘grazing of all animals’.

Participation in animal shows, pastures being in contact with the neighboring farms’ pastures, animals drinking from natural waterbodies, and employees keeping production animals at home were evaluated as binary variables (yes/no).

Univariable regression analyses were performed to identify possible risk factors for herd *CB* seropositivity. Variables with *p* < 0.35 (Appendix A) were included in the first multivariable model building. A stepwise forward inclusion procedure was used to develop the model by first adding the most biologically plausible factors and controlling for confounders and interaction terms.

The first regression model was developed to compare the prevalence of *CB*-seropositive animals in different categories of cattle herds.

The interaction terms for production type (beef/dairy) and herd size (number of animals) were added to the model because of the different sizes of the beef and dairy cattle herds and the significant effect of herd size on *CB* infection status of the herd.

The second multivariable model was built to identify the risk factors for *CB* seropositivity status of a herd among dairy cattle herds. The inclusion of variables (Appendix A) and the process of model building were the same as those described above. The keeping system, number of animals in the herd, and region were included in the final model.

Akaike information criterion (AIC) was used to monitor the fit of the models during construction. All variables with *p*-values < 0.05 were considered to have a statistically significant impact on the herds’ infection status. The analyses were performed using R version 4.1.0 [21].

## 3. Results

### 3.1. Seroprevalence Study and Risk Factor Study 1

The sample included 504 cattle herds, of which 324 were dairy and 180 were beef cattle herds. The average size of the herds was 278 for dairy herds (median 82) and 72 for beef herds (median 45). More than 90% of the dairy cattle were Estonian Holstein Friesian, Estonian Red, and/or Estonian Native cattle breeds. The beef cattle population consisted of more than 10 different breeds, of which Hereford, Aberdeen-Angus, and Limousine breeds were dominant.

The small ruminant flocks consisted of 170 sheep and 18 goat flocks. The average flock size was 165 for sheep (median 110) and 52 for goats (median 16).

The prevalence of cattle herds, sheep, and goat flocks with at least one *CB*-seropositive test result is presented in Table 1.

The apparent prevalence of *CB*-positive herds was significantly higher in dairy cattle herds than in beef cattle herds and sheep flocks. The prevalence of *CB*-positive beef cattle herds and sheep flocks did not differ significantly.

DNA of *CB* was found in 11.36% (6.29–19.67 95% CI) of the *CB*-seropositive pooled milk samples (n/N = 10/88) from the EBL surveillance program.

The results of the univariable binomial logistic regression analysis for the risk factors for cattle herds are presented in Appendix A. Risk factors associated with a cattle herd testing *CB*-positive in the univariable analysis were production type, region, number of animals, herd size, and season. The production type, number of animals, and region were included in the final multivariable model (Table 2). Cattle breeds were excluded from the final model because of evidence of collinearity with herd size. Including the interaction term for herd type (dairy or beef) and the total number of animals resulted in a slight decrease in the odds of testing positive, and was decided to retain in the model.

The final multivariable model based on samples collected via surveillance programs predicted higher odds for detecting *CB* antibodies in animals in dairy cattle herds, and if the herd was located in regions other than Southwest of Estonia (Appendix A).

### 3.2. Risk Factor Study 2 (Dairy Cattle Herds)

In total, 72 dairy farms participated in this study, submitted bulk tank milk samples for *CB* antibody testing, and replied to the questionnaire.

The results of the univariable binomial logistic regression analysis of the risk factors for dairy herds testing *CB*-positive are presented in Appendix A. The univariable analysis revealed that herds with all animals grazing, not keeping lactating cows tied, and located in the northeast region compared to those in southwest Estonia had higher odds of testing *CB* positive. Of the variables that passed the inclusion criteria for the multivariable analysis (Appendix A), three remained in the final regression model, as presented in Table 3. The keeping system was added as a fixed variable and the number of animals in the herd and location were kept in the final model as possible confounders.

## 4. Discussion

To our knowledge, this was the first study in Estonia to estimate the extent of the spread of *CB* infection in Estonian domestic ruminant herds and to identify possible associated risk factors.

The reuse of sera from the EBL surveillance program allowed taking a randomized sample of the Estonian cattle herds balanced according to the regional distribution of herds. Therefore, the obtained sample could be considered relatively well representing the target population and the prevalence estimates valid.

However, the small number of small ruminant flocks sampled in *Brucella* surveillance program did not allow to estimate reliably the herd prevalence in the population and the results reflect primarily the situation in flocks involved in breeding program, dairy flocks and larger production flocks.

The number of samples that could be obtained from goat flocks was particularly limited, and the results accordingly should be interpreted with caution.

The number of samples collected from each cattle herd was sufficient to detect a seropositive herd at defined prevalence level (design prevalence). However, the number of samples collected from small ruminant flocks might have been insufficient to detect all *CB*-seropositive flocks among the sampled ones as the within-herd prevalence may be lower than 20% in some situations [22].

### 4.1. Coxiella Burnetii Seroprevalence in Cattle Herds

*CB* antibodies were found in herds from all regions of Estonia, where *CB* infections may already be endemic. The proportion of Estonian dairy cattle herds that tested positive for *CB* was lower than that observed in Danish, Dutch, and French dairy cattle herds [10,23,24], but higher than that observed in northern parts of Europe closer to Estonia, such as Norway and Sweden [8,25]. *CB* infections have spread in different regions of France and Denmark [23,24] and are linked to abortion cases in cattle [24]. This suggests that *CB* infection is endemic in Western Europe and poses a threat to animal health. The study populations and laboratory methods used in the different studies varied, and comparisons between their results should be made with caution.

The presence of *CB* antibodies found in cattle herds indicates that cattle, as hosts of an active infection, pose a risk of spreading zoonosis to humans, such as farmers and veterinarians [26,27]. This risk has been described in human QF outbreaks associated with dairy cattle farming [28]. Since the stage of the infection has not been clarified and QF has not been diagnosed in Estonian cattle, further studies are needed to detect the dynamics of *CB* infection in cattle herds and the risk to people handling the animals.

### 4.2. Detection of Coxiella Burnetii DNA in Pooled Milk Samples

*CB* DNA has been detected in the feces, vaginal mucus, milk, and abortion material of dairy cattle, indicating an active infection [29,30]. The results of *CB* DNA-positive pooled milk samples collected for surveillance indicated that more than one out of ten herds in Estonia had at least one animal with active infection at any given time. Older animals are at a higher risk of being antibody positive to *CB* likely because of a potentially longer exposure time [31]. In 2012, the average number of lactations for dairy cows in Estonian dairy herds was 2.4 [32]. Approximately 60% of cows were in their first or second lactation cycles, whereas 22% had been in the herd for more than three lactation cycles [32]. As we could not determine the average age of animals in study herds, and young animals (calves and heifers) were not sampled, the relationship between exposure and age factors potentially influencing the onset of *CB* infection could not be determined in Estonian cattle herds. Parturition may or may not be associated with the secretion of *CB*, and the current state of knowledge suggests that control measures should focus on the whole herd, not only on infected or susceptible animals [29,33].

DNA detection in pooled milk may indicate intermittent or persistent excretion of the pathogen [29,33], which may be caused by intramammary infection [34]. Intramammary infections may result in increased environmental contamination and *CB* transmission at the conventional milking parlor, which is commonly used in Estonian dairy cattle herds (personal observation). The risk of detecting *CB* DNA in tank milk has been found to be lower in herds using automatic milking systems [10], which is attributed to differences in management practices, hygiene, behavioral patterns, and interactions between animals [35].

### 4.3. Coxiella Burnetii-Seropositivity Detection in Sheep and Goat Flocks

The study results demonstrated that the prevalence of *CB*-seropositive Estonian sheep flocks was relatively low, but it indicated that the infection occasionally occurs. Compared to Western and Central European sheep flocks, the proportion of *CB* seropositive Estonian sheep flocks was lower, similar to what has been observed in Northern Europe. A lower *CB* seroprevalence has been reported in sheep flocks in Norway and Sweden [8,25], and higher in the Netherlands, France, and Sicily, Italy [9,24,36]. *CB* has been detected in different areas of France and the Netherlands [9,24] and clinical signs of QF occur [24], suggesting that *CB* infection is endemic in sheep in these countries. *CB* antibodies have been detected in both dairy and meat production systems [9,36]. Kampen et al. [8] suggested that the low prevalence in Norwegian sheep could be explained by the limited import of animals, as domestic ruminant populations are not separated. Estonian cattle and sheep populations are usually separated, preventing spread between the species through direct contact [13]. Additionally, sheep are kept and managed similarly to beef cattle, which may partially explain the low *CB* seroprevalence (personal observation).

Because human QF outbreaks are often associated with small ruminants farming or contact with infected animals or their products, even a low *CB* prevalence in sheep flocks may pose a public health risk [3].

No *CB* antibody-positive goat flocks were detected in Estonia. This finding is similar to those reported in Norway and Sweden [8,25]. However, this study found no positive goat flocks, the number of tested goat flocks was small in this study, and the absence of the pathogen in this host could not be verified. In other parts of Europe, the prevalence is high in goats. For example, 43.1% of goat farms tested positive for *CB* antibodies in the Netherlands and 61% in France [7,24]. Kampen et al. [8] suggested that the low prevalence may be due to limited imports and combined ruminant farming. Ohlson et al. [25] reported that a low prevalence in goats coincided with a low prevalence in cattle in Sweden, indicating that ruminants may share similar risk factors for transmission.

### 4.4. Risk Factor Analysis

A higher *CB* seroprevalence was observed in dairy cattle herds than in beef cattle herds. These results are in accordance with those reported in other countries [24,31]. A similar trend has also been observed between sheep used for dairy and meat production [9,24]. A higher prevalence of *CB*-positive herds has been associated with larger herds, higher herd density of the area, managemental differences including housing practices and herd size, and outdoor and indoor environmental conditions [9,24]. Ryan et al. [37] identified dairy cattle breed as a potential animal-level risk factor, but mentioned that breed-specific effects need to be further studied, as breeds may reflect different husbandry and management techniques. In this study, specific breeds were associated with the production type and herd size and were not included in the final model.

Although human QF outbreaks are mostly associated with small ruminants [6], outbreaks can also be related with cattle [28]. In Estonia, QF is a notifiable disease in ruminants, and animals are tested when there is a clinical suspicion of the disease or due to trade requirements [38].

As this study revealed that *CB* infection is present in Estonian cattle herds, especially in dairy cattle, it is necessary to raise farmers’ awareness about the possible risks associated with *CB* infection in humans and animals. Schimmer et al. [39,40] found that working with *CB*-seropositive goats increased the odds of the farmer being seropositive as well, while no such relation was identified in dairy cattle farmers. Farmers should also be encouraged to contact veterinary authorities in case of animal health issues possibly related to QF, so that the spread of the infection can be promptly limited. Testing of imported animals originating from QF-endemic areas should be encouraged to restrict the purchase of *CB*-infected animals. In Estonia, 9.62% of veterinary professionals and 7.73% of dairy cattle farmers tested *CB* seropositive, which is higher than that in the general population (3.9%) [27]. Farmers and veterinarians thus represent a risk group for infections with *CB* and should be informed of the risks associated with working or having contact with ruminants of unknown *CB* status.

Larger cattle herds had higher odds of being exposed to *CB* in Estonia. Similarly, Spanish, Danish, and Irish cattle farms with more animals are more likely to be *CB* antibody positive [11,37,41]. These observations have also been reported in goats in the Netherlands [7] and sheep in Italy [36]. An increased number of interactions between animals may increase the possibility of contact with the pathogen from an infected animal. Differences in herd characteristics, herd management practices, and a larger number of employees and visitors to larger herds could also facilitate the introduction and transmission of infection [7,11,31,37]. Our findings support some of these findings. According to our observations, Estonian beef cattle are often raised extensively on grasslands with no or minimal natural shelter during summer and are housed in winter. Dairy cattle are often kept in intensive production systems that commonly use large indoor living areas, where *CB* can easily spread. In these farms, dairy cattle are kept loose and interact more closely with each other than beef cattle on pastures. When using indoor housing, infectious pathogens might move more easily in the environment owing to the accumulation of dust, formation of aerosols, and transmission between animals. Moreover, larger dairy herds have a number of contacts with visitors, including veterinarians, inseminators, hoof trimers, and animal husbandry consultants, who can present a risk of introducing *CB* to the farm or between animals. However, a negative association in the univariable analysis of veterinary service was detected in Denmark, presumably due to adequate personal hygiene measures [11].

Estonian cattle herds were relatively evenly distributed between the counties [13] and *CB*-seropositive cattle herds were found in all regions. Cattle from herds located in the southwestern part and dairy cattle from northwestern herds were less commonly *CB* seropositive than those from other regions of Estonia. Regional differences have been observed in other European countries. In Northern Ireland, *CB* seroprevalence in different regions ranges from 33% to 63% in cattle herds and from 3% to 10% in individual animals [31]. Significant regional differences and clusters have been reported in Swedish dairy cattle herds [25]. Production density, climate, and geographical differences have been suggested as potential factors that can affect the presence of *CB* in different regions [25,42]. In Estonia, the climate differs distinctively between the eastern and western halves of the country (e.g., the yearly average temperature in the western region is 1 °C higher than that in the eastern region). More detailed studies on regional differences in *CB*-seropositive herds are needed and can be used to conceptualize strategies for eradication programs.

The prevalence was lower in dairy cattle herds where animals were kept tied compared to herds, keeping the animals loose, or combining tied and free-range management in univariable analysis. A higher *CB* seroprevalence has also been found in Danish loose housing systems than in cattle kept tied [11]. Keeping heifers in tie stalls has significantly reduced the risk of seroconversion in Belgian dairy cattle herds, although this result was considered to be related to a low exposure level [43]. The protective effect of keeping animals tied may be due to less contact between animals shedding *CB* bacteria and susceptible animals in the herd. Furthermore, limited movement has been suggested to reduce transmission within farms [11]. However, the tied keeping of cattle is a disappearing keeping system in the European Union according to animal welfare rules for cattle and cannot be considered a measure to control infections. Farmers must find other ways to mitigate infection risk.

## 5. Conclusions

This study established serological evidence of exposure to *CB* for beef and dairy cattle and sheep, but not for goats in Estonia. Approximately one-tenth of the seropositive samples from cattle also contained *CB* DNA, indicating that a considerable proportion of exposed cattle may also shed the bacteria, and cattle may pose a risk for human infections. A larger herd size appears to be a risk factor for *CB* seropositivity in cattle herds, indicating that certain management practices may play a role in *CB* spread. Regional differences in CB seropositivity in cattle herds may reflect the effect of climatic conditions on *CB* spread in Estonia. The results of this study can be used to inform risk groups for zoonosis and to implement control strategies to limit the spread of *CB* in ruminant herds.

## Figures and Tables

**Table 1 microorganisms-11-00819-t001:** Prevalence of *Coxiella burnetii* antibody-positive herds among beef and dairy herds and sheep and goat flocks sampled for national surveillance programs in Estonia during 2012–2013.

Ruminants	n/N ^1^	Prevalence, % (95% CI ^2^)
Beef cattle herds	12/180	6.67 (3.85–11.29)
Dairy cattle herds	88/324	27.16 (22.61–32.25)
Sheep flocks	4/170	2.35 (0.92–5.89)
Goat flocks	0/18	0.00 (0.00–16.82)

^1^ number of herds (n) with at least one *CB*-seropositive test result and number of tested herds (N). ^2^ 95% confidence intervals.

**Table 2 microorganisms-11-00819-t002:** Binomial logistic regression analyses of risk factors associated with *Coxiella burnetii*-seropositive cattle herds sampled for national surveillance programs in Estonia during 2012–2013.

Variable	n/N ^3^	Odds Ratio (95% CI ^4^)	*p*-Value	ANOVA *p*-Value
Production type				
Beef cattle	12/180	1		<0.001
Dairy cattle	88/324	5.40 (2.07–14.04)	<0.001	
No. of animals ^5^		1.007 (1.001–1.013)	0.016	<0.001
Region				
Southwest	14/156	1		<0.001
Southeast	25/99	3.80 (1.75–8.22)	<0.001	
Northeast	40/121	4.60 (2.24–9.41)	<0.001	
Northwest	21/128	2.46 (1.13–5.39)	0.024	
Interaction ^6^				
Beef cattle herds: No. of animals		1		0.114
Dairy cattle herds: No. of animals		0.99 (0.99–1.00)	0.100	

^3^ number of herds (n) with at least one *CB* antibody positive result and number of tested herds (N); ^4^ 95% confidence intervals (95% CI); ^5^ total number of animals in the herd (No. of animals); ^6^ interaction between production type and total number of animals in the herd

**Table 3 microorganisms-11-00819-t003:** Multivariable binomial logistic regression analyses for the risk factors associated with dairy cattle herds testing seropositive for *Coxiella burnetii* in a bulk tank milk sample in Estonia.

Variable	n/N ^7^	Odds Ratio (95% CI ^8^)	*p*-Value	ANOVA *p*-Value
Keeping system (lactating cows)				
Tied	5/36	1		0.087
Mixed ^9^	3/5	3.54 (0.28–45.64)	0.332	
Loose	17/31	5.56 (1.17–26.39)	0.031	
No. of animals ^10^		1.001 (1.000–1.002)	0.359	
Region				
Southwest	8/25	1		0.002
Southeast	3/13	0.50 (0.07–3.30)	0.468	
Northeast	13/18	3.29 (0.71–15.17)	0.127	
Northwest	1/16	0.09 (0.01–0.92)	0.042	

^7^ number of herds with at least one *CB* antibody positive result (n) and number of tested herds (N); ^8^ 95% confidence intervals; ^9^ loose and tied keeping systems used; ^10^ number of animals in the herd.

## Data Availability

Data are available on request.

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
