# Peer review of "Coxiella burnetii Seroprevalence and Associated Risk Factors in Cattle, Sheep, and Goats in Estonia"

_microorganisms, 2023, doi:10.3390/microorganisms11040819_

Round 1

Reviewer 1 Report

The manuscript is about the prevalence of Coxiella burnetii antibodies in Estonian ruminants, and associated risk factors. 

Please check for spaces and full stops, italics, etc. in the manuscript.

The methodology is a bit confusing. Samples were collected in 2012 and 2013, how can authors justify the prevalence of CB  in stored samples over a decade? 

Please give the detail of the sample storage conditions.

At what temperature the authors stored the samples?

When DNA was extracted (year)??

Please add a reference in the antibody detection section (line 127).

In the DNA detection section, please provide the detail of primers, and thermocycler conditions.

Did you use positive and negative control in the experiment? Please provide detail.

How many samples were used for DNA extraction from different ruminants? 

Please modify the methodology to make it reader-friendly and comprehensive.

Line 244-245: "To our knowledge, this was the first study in Estonia to estimate the extent of the spread of CB infection in Estonian domestic ruminant herds and to identify possible associated risk factors". 

However, you have mentioned in the introduction (lines 53-55), you wrote:

"In Estonia, there are few data on the prevalence of CB in domestic animals. Evidence of infection has previously been detected serologically in five cattle from three different farms [12]". Please justify. 

Discussion (lines 247-256): I am not sure what the authors intended to discuss. 

Line 273: Coxiella burnetii should be italicized.

Line 280, 292, 299, 307, 309, 312, 322, 324, 363, 367: Add references.

The authors should work on the discussion to make it accurate, succinct, and to the point.

Please write the conclusion in one paragraph.

Please proofread the manuscript to correct English errors.

Author Response

Dear Reviewer 1

We are very grateful for your time and effort in improving our article before publication.

Please see the added Word document to view our responses to your questions and comments.

Sincerely yours,

Kädi Neare (in the name of all authors)

Reviewer 2 Report

The manuscript presents  very interesting results regarding the seroprevalence and risk factors to Q fever in Estonia. Sampling and data analysis  are described in detail providing an excellent contribution to the knowledge of Coxiella burnetii in this study area.

Author Response

Dear Reviewer 2,

Our research group is very grateful for your time and effort in reviewing the submitted article. We thank you for the kind review and wish you all the best for the future.

Sincerely yours,

Kädi Neare (in the name of all authors)

Reviewer 3 Report

Dear authors,

The present manuscrip is a decent attempt to describe the epidemiology Coxiella burnetii in Estronia.

I have highlited some minors syntax errors which unfortunately make the work seem sketchy.

Since figure S1 is missing I did not understand the justification of sample size in sheep and goat. How many samples did you analized? Were selection criteria applied to sheep and goats?

Accroding to the text "The number of samples collected from each flock enabled the identification of an infected flock with 95% confidence, at least on a 20% prevalence level, assuming 88.8% sensitivity and 105 98.5% specificity of the test in sheep, and 91.6% sensitivity and 98.9% specificity in goats,respectively" . Are these precentages too low to give robust conclusions? Βy choosing a hypothetical or estimated  prevalence, wouldn't it be better to show us the power of the study?

Author Response

Dear Reviewer 3,

Our research group is very grateful for your time and effort in reviewing the submitted article.

Please see our responses to your comments and questions in the added Word document.

Sincerely yours,

Kädi Neare (in the name of all authors)

Round 2

Reviewer 1 Report

The authors complied with almost all the comments and improved the methods and discussion. 

Just minor comments.

Please add a reference at the end of the DNA detection section.

I am curious why the authors did not do the phylogenetic analysis for CB in Estonia. However, they can do this in their future studies to detect CB strains. 

The overall manuscript is improved. 

Author Response

Dear Reviewer 1,

Our research group is very grateful for your time and effort in reviewing the submitted manuscript. Please see our responses to your latest comments and notes (added below).

  • Comment: Please add a reference at the end of the DNA detection section.

Response: Thank you for letting us know about the missing reference. Now it has been added.

  • Comment: I am curious why the authors did not do the phylogenetic analysis for CB in Estonia. However, they can do this in their future studies to detect CB strains. 

Response: The phylogenetic analysis was planned to be a part of the study and in relation, in 2014 the MLVA analysis was performed on all milk samples containing CB DNA. The results of these analyses are planned to be published as part of our next paper.

We thank you for the kind review and wish you all the best for the future.

Yours faithfully,

Kädi Neare (on behalf of all authors)